# Does the Exchange Rate and Its Volatility Matter for International Trade in Ethiopia?

**Tiblets Nguse [1], Betgilu Oshora [1]** ![], **Maria Fekete-Farkas [2]** ![], **Anita Tangl [3]** ![] **and Goshu Desalegn [1,*]** ![]

1   Doctoral School of Economics and Regional Sciences, Hungarian University of Agriculture and Life Sciences, Páter Károly u. 1, 2100 Gödöllő, Hungary; Baraki.Tiblets.Nguse@hallgato.uni-szie.hu (T.N.); olle.betgilu.oshora@phd.uni-szie.hu (B.O.)
2   Institute of Agricultural and Food Economics, Hungarian University of Agriculture and Life Sciences, Páter Károly u. 1, 2100 Gödöllő, Hungary; farkasne.fekete.maria@uni-mate.hu
3   Szent István Campus, Institute of Rural Development and Sustainable Economy, Hungarian University of Agriculture and Life Sciences, Páter Károly u. 1, 2100 Gödöllő, Hungary; tangl@vajna.hu
*   Correspondence: Deresa.Goshu.Desalegn@phd.uni-mate.hu

**Abstract:** This study was carried out to investigate the impact of the Ethiopian exchange rate and its volatility on international trade. Trade openness was used as a proxy for international trade in the study. The study's general objective was to investigate how international trade responds to exchange rate levels and volatility. The study relied solely on secondary time-series data spanning the years 1992 to 2019. The Autoregressive Distributive Lag (ARDL) model was used in the study to investigate the long-term relationship between exchange rate level, volatility, and international trade performance. An error correction model was used to estimate the variables in the short term. To conduct the regression analysis, Foreign Direct Investment (FDI), Gross Domestic Product (GDP), and inflation were used as control variables. The finding of the study implies that: in the short term, the exchange rate level was found to negatively and significantly influence international trade. However, exchange rate volatility positively and significantly affects international trade both in the short and in the long term. In addition, gross domestic product, foreign direct investment, and inflation have a positive effect on international trade both in the short term and long term. This finding lends support to the J-curve effects, which suggest an initial loss in the short term followed by a dramatic gain in the long term. However, the findings of this study suggest that there is no significant gain from international trade to justify currency depreciation in Ethiopia.

**Keywords:** exchange rate; exchange rate volatility; international trade

## 1. Introduction

The foreign exchange rate is one of the most important factors in determining a country's relative economic growth. The rate at which one country's currency is transformed into another is known as the exchange rate. The foreign exchange rate of a country is a new window into its economic stability (Qin 2000). The exchange rate is critical in controlling the home economy's broad allocation of production and consumption between foreign and domestic commodities.

The exchange rate level is the most significant predictor of the export level, according to Uusivuori and Laaksonen-Craig 2001, and export-led growth has received special attention in several nations. This is due to the fact that exports create limited foreign exchange reserves, which are required to fund critical imports for domestic production and capital formation. Increased export earnings can also help a country's balance of payments. Furthermore, the reinvestment of export revenues could play a significant role in increasing job prospects for the unemployed (Baek and Okawa 2001).

In this case, a high exchange rate level reduces the receipts received by exporters, lowering export earnings. A low exchange rate level, on the other hand, increases receipts

received by exporters, thus increasing export earnings. Exchange rate fluctuations may have a negative impact on exporters as well as economic growth by discouraging firms from investing, innovating, and trading. It may also discourage businesses from entering the export market. A study by (Barguellil et al. 2018) argues that volatility seems to be more harmful under a flexible exchange rate regime and financial openness for developing and emerging countries. International trade revenues are generally paid in any of the hard currencies, which are then transferred to the local currency to facilitate and pay local responsibilities (Bende-Nabende 2002).

The tendency for foreign currencies to appreciate or depreciate, affecting the profitability of foreign currency trades, is referred to as exchange rate volatility. Volatility is defined as the amount of variation in these rates as well as the frequency with which they fluctuate (Ozun 2007). It is claimed that countries that devalue their currencies have the highest levels of exports.

"Since 1990, numerous studies by different scholars such as (Thorbecke and Kato 2012), (Mbam and Michael 2020) and (Oloyede and Isaac 2017)" have been conducted to examine the effect of exchange rate and exchange rate volatility on international trade in general. However, the results of the studies lack consistency from country to country. Furthermore, evidence from the literature drawing on a range of countries over time tends to suggest that it is important to keep the exchange rate expensive though not necessarily undervalued (Ferrand 2018). In this case, all conducted studies and established theories ended up with different results in different countries. Despite ambiguous empirical findings, many developing countries have continued to use currency depreciation as a growth strategy (Ferrand 2018).

In context of Ethiopia, since the implementation of the floating exchange rate in the early 1990s, the Ethiopian Birr has been steadily depreciating. As an example, consider the recent devaluation of the US dollar in 2017 (International Monetary Fund (IMF) (2018)). The IMF estimated in 2015 that the birr was 30% overvalued. This overvaluation was exacerbated by the country's structural trade deficit, which should have depreciated its currency mechanically (Wondemu and Potts 2016). The result of this overvaluation was a competitiveness deficit for exports, which were more expensive than they should have been and thus less appealing. The overvalued birr, on the other hand, lowers the cost of imports. This is an understandable choice for a country with a large trade deficit.

The government took action in 2017 to depreciate the birr currency rate by 15%. Economic players see the measure as helping to boost the growth of the country's export sector, which has seen a slowing outlook (International Monetary Fund (IMF) (2018)). It was also expected to relieve debt and reduce currency shortages. Although a high return on investment was expected to prevent inflation as a result of the devaluation, this did not happen. Domestic output must be responsive to meet existing and rising demand, which is caused by the birr's depreciation, for devaluation to be successful. If demand for Ethiopian exportable goods increases, there must be excess or spare capacity. However, the devaluation measure will raise the cost of production for state-owned, private, and even partially-owned enterprises that rely heavily on imported intermediate/capital goods (inputs), reducing their capacity utilization capabilities. It is also possible that other trading partners follow suit (i.e., devalue their currency so that Ethiopia cannot take advantage of them) or take other retaliatory measures. Furthermore, there is uncertainty about the availability of domestically (Ethiopian) produced goods that both domestic and foreign consumers wish to purchase.

If Ethiopia, which now has a weaker birr, does not reduce imports, it will require more money to pay for the same amount of foreign goods (Bekele 2019). In this case, the measure will be ineffective in addressing Ethiopia's serious trade imbalance. Others, on the other hand, bring the traditional theory of the J-curve into play, arguing that, while the birr's depreciation may worsen the country's current balance of payments position in the short term, the devaluation measure may lead to improved trade balances in the long term.

With this in mind, and the influence of the Ethiopian government making to initiate international trade by using exchange rate policy as one available tool, this study mainly focuses on essentially what the effect of the exchange rate level and exchange rate volatility have on international trade. Hence, the main objective of this study is to examine the effect of the exchange rate level and its volatility on international trade in Ethiopia. In doing so, the study used foreign direct investment, economic growth, and inflation as control variables for the study. The paper is structured in five sections: The next section reviews the literature and the econometric theory involved in the past and present studies. Section 3 describes the research methodology applied in the study, while Section 4 is discusses the results. The last section offers the conclusion and policy recommendation.

## 2. Literature Survey

To be involved in the international market, one currency has to be expressed in terms of another currency through the exchange rate. The exchange rate is expressed as the price of one currency in terms of another (Mishkin and Eakins 2009). An exchange rate can be explained as either a direct or indirect quotation. A direct quotation denotes how much of the home currency can be used to purchase a unit of the foreign currency, whereas an indirect quotation denotes how much of the foreign currency can be obtained from a unit of the home currency (Frijns 2015). When inflationary effects are included, the exchange rate is referred to as a nominal exchange rate; when inflationary effects are excluded, the exchange rate is referred to as a real exchange rate (Lothian and Taylor 1997).

Following the 1970s and 1980s, many countries, particularly those in the Third World, saw currency depreciation, or the depreciation of their own currency in terms of foreign currencies, as a critical issue for economic growth. In this case, as confirmed by many studies, the exchange rate plays a significant role in deciding the level of international trade. The exchange rate can be adjusted through depreciation and appreciation of local currency against foreign currency, which has huge implications for multinational companies (Xing and Zhao 2008). However, currency fluctuations can have an expansionary or contractionary impact on economic growth. Many development organizations, such as the International Monetary Fund (IMF), support currency depreciation as a means of boosting economic growth, in addition to the financial aid and loans they provide to their member countries for the development of domestic firms. Hence, devaluation is expected to boost firm competitiveness and increase domestic product and output production. Some researchers, however, are not agreed on these, as (Fonchamnyo and Akame 2017) shed light on the negative effects of currency depreciation on output.

On the other hand, exchange rate volatility may occur due to unexpected movements in the exchange rate and higher exchange-rate volatility leads to higher costs for risk-averse traders and less foreign trade. If changes in exchange rates become unpredictable, this creates uncertainty about the profits to be made and, hence, reduces the benefits of international trade (Hook and Boon 2000). The empirical study result by (Shevchuk and Kopych 2021) in central and eastern European countries reveals that exchange rate volatility reduced the risk of recession in the Czech Republic, while the opposite effect was found for Hungary and Romania, with neutrality for Poland.

A currency devaluation is a deliberate act by a country's central bank to reduce the value of its currency in relation to other countries' currencies. In theory, currency depreciation is a tool for improving the economy's export sector. Devaluation raises the price of a country's imports relative to its exports; as a result, exporters earn more domestic currency from a given export quantity, while imports contract due to the higher domestic currency price of imports. Thus, depreciation functions similarly to a tax on imports and a subsidy on exports, causing the trade balance to improve (Kandil 2008). At the same time, trade policy may be used to mitigate the effects of an overvalued currency.

Domestic firms that lose competitiveness as a result of an increase in the real exchange rate may lobby for trade restrictions. In practice, disagreements among trading partners over exchange rate policies may lead to an increase in domestic political pressures and

unilateral trade action (Marjit and Ray 2021) In a broader sense, countries may use trade policy to compensate for exchange rate overvaluation in order to address persistent trade imbalances.

The aggregation problem could also be a contributing factor to such contentious results. The effects of the exchange rate on export and foreign direct investment may differ across industries. Xu and Guo (2021) suggest that this is possible because the level of competition, the price-setting mechanism, currency contracting, the use of hedging instruments, the economic scale of production units, openness to international trade, and the degree of homogeneity and storability of goods differ across sectors. In case of developing countries, differences in exporters' access to financial instruments, currency contracting, production scale, storability, and so on may be more pronounced. This disparity is exacerbated by the fact that agriculture is typically a highly competitive industry with flexible pricing on relatively short-term contracts (Rahmati et al. 2021).

Furthermore, the study conducted by (Bahmani-Oskooee and Brooks 1999), shows the relationship between the trade balance and its determinants relied on disaggregated bilateral data from the USA and six of its largest trading partners and implied that the trade balance has a short and long term response to currency depreciation. More specifically, the result of the study showed no specific short-term pattern supported the J-Curve phenomenon, as the long-term results did. The study implies that the long-term results supported the economic theory, indicating that a real depreciation of the dollar has a positive long-run effect on the USA's trade balance with her six trading partners.

To this end, the study conducted by (Bahmani-Oskooee and Ratha 2004) showed how using different data (aggregate and disaggregate) affected the relationship between trade balance and exchange rate level. The study further inferred that data usage for testing the relationship between the variables had a significant impact on the result of the studies. The study's findings imply that the short-term response of the trade balance to currency depreciation did not follow any particular pattern. However, when it came to the long-term effects of depreciation, models that relied on bilateral trade data produced more results that supported the positive long-term relationship between exchange rate and trade balance when compared to aggregate data.

In addition, (Bahmani-Oskooee and Wang 2008) conducted a study to show the relationship between the trade balance and exchange rate level at industry level by taking a case of the USA and China trade partners (disaggregated by commodity) in testing (J-curve effect) and the long-term effects of currency depreciation on the trade balance. The imports and exports of 88 industries were used in the study, and cointegration analysis was used to demonstrate their relationship. The study's findings implied that the trade balances of at least 34 industries responded favorably to real depreciation of the dollar. The J-curve effect was observed in 22 industries. The study conducted by (Lucarelli et al. 2018) showed how Euro depreciation and trade asymmetries between Germany and Italy versus the USA affected industry-level estimates. The study was conducted using industry-level data at monthly frequency. The outcomes of the study differed depending on the bilateral relationship. Their find implies that for the given industries, 11 industries improved in the long run (eight for Italy and three for Germany) because of the euro depreciation. According to the finding of the study, the J-curve effect has only been demonstrated in six cases, all of which involve Italy. However, the study also demonstrated the inverted J-curve effect in eight industries, four in Germany and four in Italy. These findings imply that there will be different responses to currency depreciation at the industry level. To support this, Gobbi and Lucarelli (2021) conducted study to show Euro depreciation and supply chains response to industry-level estimates for Germany, Italy, and Greece. According to the empirical findings, the Euro depreciation increased the integration of German and Greek production structures in various industries, accounting for more than 35% of total trade between the two countries. In particular, the contributions of Lucarelli et al. (2018) and Gobbi and Lucarelli (2021) point out that looking at different industries even within

the same country there can be different effects of currency depreciation on international trade.

Specifically, at the sectoral level, (Chebbi and Olarreaga 2019) tried to investigate the impact of changes in Tunisia's exchange rate on the net external position of the agricultural sector. Their findings showed that the long-term and short-term impact of exchange rate changes on the net agricultural trade balance, and that depreciation of the domestic currency led to a deterioration of Tunisia's agricultural sector's net external position in the long run. Keho 2021 examined the determinants of the trade balance in the West African and Monetary Union (WAEMU) from 1975 to 2017. The findings showed that the trade balance was negatively related to domestic and foreign income, whereas real effective exchange rate deprecation improved trade balance in the long run. However, the findings did not support the J-curves of a short-term worsening of trade balance. In the short term, the trade balance was only sensitive to foreign real income and not to domestic income or the real exchange rate. Apart from the exchange rate, there are other determinants which affects the level of trade.

(Lakew 2003), a national bank of Ethiopia staff member, investigated the main determinants of the country's exports, on the one hand, and highlighted the opportunities available both at home and abroad, as well as the challenges that the country's export sector faced in today's globalizing and integrated world, on the other. The study's findings implied that the real exchange rate, real private sector credit, and real private consumption were significant long-term determinants of the country's exports. In the short term, the main export determinants were real GDP, real private sector credit, and real private consumption.

To this end by taking into account the importance of the subject, many studies have been conducted to examine the issue in general, as well as in particular (Ullah et al. 2012), (Aizenman et al. 2012), (Kiyota and Urata 2004), (Babecký et al. 2012), (Woldekidan 1992; Lipsey 1991; Giese et al. 1990; Gopinath et al. 1998; Kulatilaka and Kogut 1996; López and Thomas 1990; Mengisteab 1997; Elbadawi 1999; Grosse and Trevino 1996; Dodsworth 1996; Rasiah 1998; Fry 1996; Bayoumi and Lipworth 1998; Kyrkilis et al. 1998; Goldberg and Kolstad 1995; Asiedu and Lien 2004; Havlik 2000; Fawaz et al. 2001; Tomšík 2001; Onishi 2002; Qin 2000; Uusivuori and Laaksonen-Craig 2001; Baek and Okawa 2001); however, they end up with controversial results. The level of export in Ethiopia fluctuates, with ups and downs. In fact, no one knows why these ups and downs occur, but it is possible that the level of the exchange rate is a variable in determining a country's trade level. The following Table 1 shows Ethiopian export performance over the past three years (2018–2020).

**Table 1.** Unit Value of Major Export Items. (In USD/kg unless stated otherwise).

| Particulars | 2017/18 | 2018/19 | 2019/20 | Percentage Change | |
| --- | --- | --- | --- | --- | --- |
| | A | B | C | B/A*100 | C/B*100 |
| Coffee | 3.52 | 3.31 | 3.16 | −10.24 | −4.59 |
| Oilseeds | 1.22 | 1.49 | 1.46 | 20.04 | −2.20 |
| Products | 20.69 | 20.99 | 20.28 | −1.95 | −3.40 |
| Pulses | 0.62 | 0.59 | 0.66 | 7.84 | 12.74 |
| Meat and meat products | 5.10 | 5.00 | 5.26 | 3.22 | 5.14 |
| Fruits and Vegetables | 0.33 | 0.35 | 0.31 | −5.33 | −11.22 |
| Textile and textile products | 6.19 | 7.69 | 7.41 | 19.58 | −3.62 |
| Live animals | 1.91 | 1.88 | 1.84 | 3.81 | −2.17 |
| Chat | 5.60 | 5.67 | 5.68 | 1.46 | 0.18 |
| Gold (In USD/grams) | 35.51 | 34.12 | 59.12 | 66.48 | 73.28 |
| Flower | 4.56 | 4.44 | 4.47 | −1.94 | 0.84 |
| Electricity (In USD/kwh) | 0.06 | 0.06 | 0.06 | 3.72 | 0.81 |

Source: National bank of Ethiopia (NBE 2020).



## 3. Methodology

In this study, the researchers used a quantitative research approach. The use of this method is helpful to ensure that the data collected are effectively interpreted and analyzed using statistical analysis and descriptive statements. Furthermore, the study used an explanatory research design to examine the effect of exchange rate and its volatility on trade openness. Based on past empirical studies the following research hypotheses were formulated for the study.

**Hypothesis 1 (H1).** *The exchange rate level has a positive and significant effect on international trade.*

**Hypothesis 2 (H2).** *The exchange rate volatility has a negative and significant effect on international trade.*

**Hypothesis 3 (H3).** *Foreign direct investment has a positive and significant effect on international trade.*

**Hypothesis 4 (H4).** *Economic growth has a positive/negative and significant effect on international trade.*

**Hypothesis 5 (H5).** *Inflation has a negative and significant effect on international trade.*

As shown in Figure 1, the sampling frame is based on time series annual data of the dependent (trade openness) and independent (exchange rate, exchange rate volatility, economic growth rate, inflation rate, and foreign direct investment) variables between 1992 and 2019. The sample period includes 28 years of annual observation of all variables. This period was sampled based on available data for exchange rates. To the best of our knowledge, there were no reliable and organized data relating to the subject of the study. Moreover, several scholars' findings were reviewed from different websites, publications, and annual reports to triangulate the present finding with the previous results.

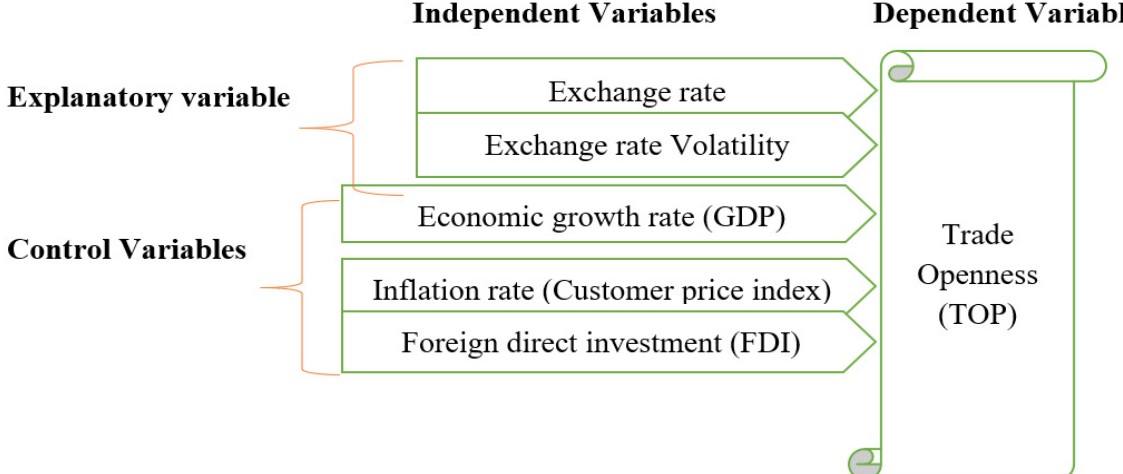

**Figure 1.** Conceptual framework elaborated by the authors.

### 3.1. Model Specification of the Study

To investigate the effect of the exchange rate and its volatility on international trade in Ethiopia, the following estimated equation was used. For the usefulness of the interpretation, the natural logarithm was used for some variables.

$$\text{TOP} = f(\text{ER, ERV, GDP, INF, FDI}) \tag{1}$$

TOP = Trade openness as % of GDP = (Export + Import/GDP). The natural logarithm is applied for this variable;
ER = Exchange rate level (USD/ETB). The natural logarithm is applied for this variable;
GDP = GDP Growth rate;
INF = Inflation rate (customer price index);
ERV = Exchange rate volatility, measured as the standard deviation of the exchange rate. The natural logarithm is applied for this variable;
FDI = Net FDI inflows. The natural logarithm is applied for this variable.

### 3.2. Model of the Study

For econometric analysis purposes, the above equation has been changed to logarithmic format to generate Equation (2):

$$LTOPt = \beta 0 + \beta 1\ LERt + \beta 2\ LERV + \beta 3 GDPt + \beta 4 INFt + \beta 5\ LFDI\ t + c \tag{2}$$

#### 3.2.1. Discussion of Model Used

In this study, the researcher used the Autoregressive Distributed Lag (ARDL) Model. The reason behind using the autoregressive distributed lag model in this study is that the series is a combination of I(0) and I(1).

#### 3.2.2. Representation of General Autoregressive Distributed Lag (ARDL) Model

The general representation of the Autoregressive Distributed Lag (ARDL) model can be written as:

$$Yt = \mu oi \sum_{i=1}^{p} \alpha jyt - 1 + \sum_{i=1}^{q} \beta jXt - 1 + \varepsilon it \tag{3}$$

$Yt$ is a vector, $\mu oi$ is the intercept, and variables in $Xt$ are allowed to be purely I(0) or I(1) or fractionally integrated; $\beta$ and $\alpha$ are coefficients; and $j = 1, \ldots k$ is several independent variables. $P$ is the lag length of the dependent variable and q is the optimal lag for independent variables, while the term $\varepsilon it$ represents a vector of error terms.

To investigate the existence of cointegration among variables, the researcher used the bounds testing approach developed by (Pesaran et al. 1999). The stationarity levels of the series are analyzed before the bound test is carried out. The following Table 2 of the study showed the result of stationarity test.

**Table 2.** Result of Stationarity test with the ADF test.

| Variables | At Level INTERCEPT | At 1st Difference INTERCEPT | Decision |
|---|---|---|---|
| LTOP | 0.1167 | 0.0001 ** | Stationary at I(1) |
| LER | 0.9958 | 0.0000 *** | Stationary at I(1) |
| LERV | 0.8751 | 0.0142 ** | Stationary at I(1) |
| GDP | 0.0001 *** | 0.0000 *** | Stationary at I(0) |
| INF | 0.0111 ** | 0.0000 *** | Stationary at I(0) |
| LFDI | 0.0039 ** | 0.0001 *** | Stationary at I(0) |

Source: the researcher's own computation using E-view 10 software. Note: *** shows stationarity of variables at 1 percent significance level, ** shows 5 percent significance level.

#### 3.2.3. Unit Root Test

Under the unit root test, several tests are available but the most commonly used is the Augmented Dickey-Fuller (ADF) test to check the stationarity of the variables. The hypotheses of these tests are also stated as:

HO:                          Unit root in variables                          decision criteria;
H1:                          No Unit root in variables;                          reject HO; if PV < 0.05.

After determining the stationarity level of the series, the existence of cointegration among the series was tested. The bounds testing approach suggested by (Pesaran et al.

1999) was used to check if cointegration existed among variables. If cointegration was found among variables, the long-term and short-term estimations of the variables were made. The following bound test was conducted to check if the long relationship among variables existed. The following Table 3 of the study showed the result of bound test.

**Table 3.** The result of the bound test.

| F-Bound Test | | | | |
|---|---|---|---|---|
| Test Statistic | Value | Significance | I (0) | I (1) |
| F-statistic | 14.02341 | 10% | 2.08 | 3 |
| | 5 | 5% | 2.39 | 3.38 |
| | | 2.5% | 2.7 | 3.73 |
| | 5 | 1% | 3.06 | 4.15 |
| | | | Asymptotic, n = 100 | |

Source: Constructed by the researchers using E-view 10.

***ARDL model estimated to test for cointegration among the variables.***

$$
\begin{aligned}
\Delta LTOPt = \; & \beta 0 \; + \; \beta 1 \, LTOPt - 1 \; + \; \beta 2 LERt - 1 \; + \; \beta 3 LERVt - 1 + \; \beta 4 GDPt - 1 \; + \; \beta 5 INFt - 1 \; + \; \beta 6 LFDIt - 1 \\
& + \; \sum_{a}^{h} \lambda 2 \, \Delta \, LER \, t \; - a \; + \sum_{b}^{h} \lambda 3 \Delta LERVt - b \; + \sum_{c}^{h} \lambda 4 \, \Delta \, GDPt - c \; + \; \sum_{d}^{h} \lambda 5 \, \Delta INFt - d \\
& + \; \sum_{e}^{h} \lambda 6 \, \Delta LFDIt - e \; + \; \varepsilon t \ldots
\end{aligned}
\tag{4}
$$

According to the bounds test, the calculated F-statistic was above the upper critical bound values (higher than at 90%, 95%, 97.5%, and 99% upper bounds), which means that the model rejected the null hypothesis of no level effects. And this implies, there was cointegration between the variables. The existence of cointegration among variables revealed the study should have to estimate with an error correction model to know the short-term relationship among variables.

***The error correction model was formulated to show the short-term relationship among the variables.***

$$
\begin{aligned}
\Delta \, LTOPt = \beta 0 \quad & + \beta 1 \, \sum_{a}^{h} i(LTOPt - 1) + \beta 2 \, \sum_{b}^{h} i(LERt - 1) + \beta 3 \sum_{c}^{h} i(LEVt - 1) \\
& + \beta 4 \sum_{d}^{h} i(GDPt - 1) + \beta 5 \sum_{e}^{h} i(INFt - 1) + \beta 6 \sum_{f}^{h} i \, (TOPt - 1) + \mu ECM \, (-1) \varepsilon t \ldots
\end{aligned}
\tag{5}
$$

Several tests were carried out in the study to determine whether the model was visible and useful for policy recommendations. To begin, a multicollinearity test was performed using a correlation matrix to determine whether or not there was a problem with the variables. Other tests, such as the normality, heteroscedasticity tests, and Ramsey reset test, were then used to confirm that the model was feasible. The test results are presented in the Appendix A.

## 4. Discussion

Unquestionably, a thorough understanding of the data is very important before embarking on econometric analysis. Therefore, different tools of descriptive statistics such as measures of central tendency, graphs, and charts were employed to check the properties of the variables. All these tools helped to identify the characteristics of the variables over the research period. Moreover, a comprehensive observation of the data helps to make a meaningful interpretation of the econometric results. Bearing this in mind, the following section elaborates the detailed information of each variable, which includes the mean, median, minimum, maximum, and standard deviation calculated using the E-views 10 software package. The following Table 4 of the study showed the result of descriptive statics for each variable.

**Table 4.** Descriptive analysis result.

|            | TOP        | ER        | ERV        | FDI      | INF        | GDP        |
|------------|------------|-----------|------------|----------|------------|------------|
| Mean       | 0.899153   | 1.025383  | 0.455873   | 8.366457 | 9.975620   | 7.494283   |
| Median     | 0.894129   | 0.938645  | 0.485045   | 8.452362 | 8.302638   | 9.118018   |
| Maximum    | 1.539972   | 1.463441  | 1.295743   | 9.617308 | 44.39128   | 13.57260   |
| Minimum    | −0.071362  | 0.447546  | −0.540649  | 5.230449 | −8.484249  | −8.672480  |
| Std. Dev.  | 0.445515   | 0.250221  | 0.550727   | 0.981870 | 10.75847   | 5.390044   |
| Observations | 28       | 28        | 28         | 28       | 28         | 28         |

Source: Researcher's own computation by taking row data from the World Bank.

### 4.1. Trend, and Descriptive Analysis of the Variables

The next part of this article presents the trend analysis of the components of trade openness, exchange rate, and exchange rate volatility.

#### 4.1.1. Trade Openness

This study was conducted by using 28 years of annual data observations from 1992 to 2019. The dependent variable in this study was trade openness, measured as a percentage of GDP. As we can see from the above descriptive analysis Table 5, the average result of trade openness as a percentage of GDP during the study period was 0.899 percent with a maximum of 1.54 percent and a minimum of −0.07 percent. This implies that during the study period, the trade openness as a percentage of GDP in Ethiopia ranged from −0.07 to 1.54 with an overall average of 0.90 percent. In addition to this, each observation in the study deviated from this average by the value of 0.44 percent. The following Figure 2 of the study showed trend analysis of trade openness in Ethiopia.

**Table 5.** Long-term estimation of the model (trade openness).

| Levels Equation | | | | |
|---|---|---|---|---|
| **Case 2: Restricted Constant and No Trend** | | | | |
| Variable | Coefficient | Std. Error | t-Statistic | Prob. |
| LER       | 0.140920   | 0.310887   | 0.453283    | 0.6584 |
| LERV      | 0.225485   | 0.065867   | 3.423310    | 0.0050 |
| LFDI      | 0.246034   | 0.052221   | 4.711367    | 0.0005 |
| INF       | 0.016336   | 0.002992   | 5.459149    | 0.0001 |
| GDP_RATE  | 0.016318   | 0.004004   | 4.075049    | 0.0015 |
| C         | −1.584880  | 0.234443   | −6.760196   | 0.0000 |

Source: Result generated by authors from E-views 10 software.

#### 4.1.2. Exchange Rate and Exchange Rate Volatility

As mentioned in the conceptual framework of this study, the explanatory variable used in this study was the exchange rate, which is measured by local currency against USD and exchange rate volatility measured by the standard of deviation of the exchange rate; the average value of this variable was 1 percent, which implies that is the annual average change to buy USD dollars. The maximum and minimum value of this variable was 1.46 and 0.44 percent, respectively, during the study period. For each observation in this study, there was a deviation of 0.25 percent from its average for the variable exchange rate. The other variable used to determine international trade was exchange rate volatility, which is measured by the standard deviation of the exchange rate. During the study period, the average score of this variable was 0.46 percent with a maximum and minimum of 1.3 percent and −0.54 percent, respectively. The following Figures 3 and 4 of the study showed the trend of the exchange rate and exchange rate volatility in Ethiopia.

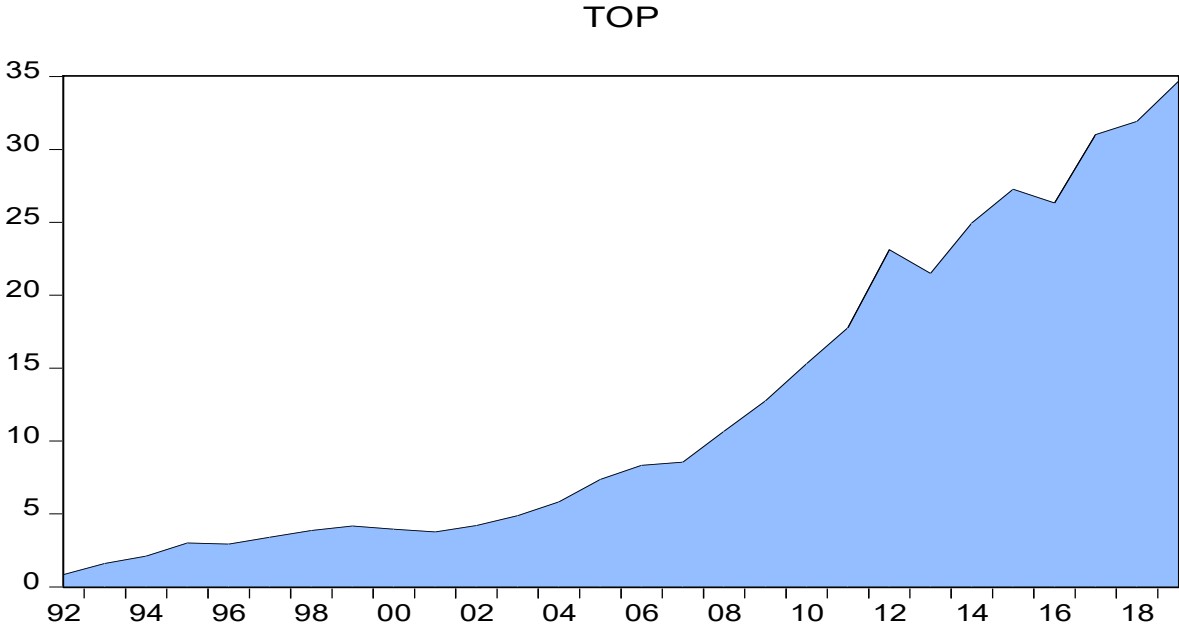

**Figure 2.** Trade openness movement in Ethiopia. Source: Constructed by authors using E-view 10.

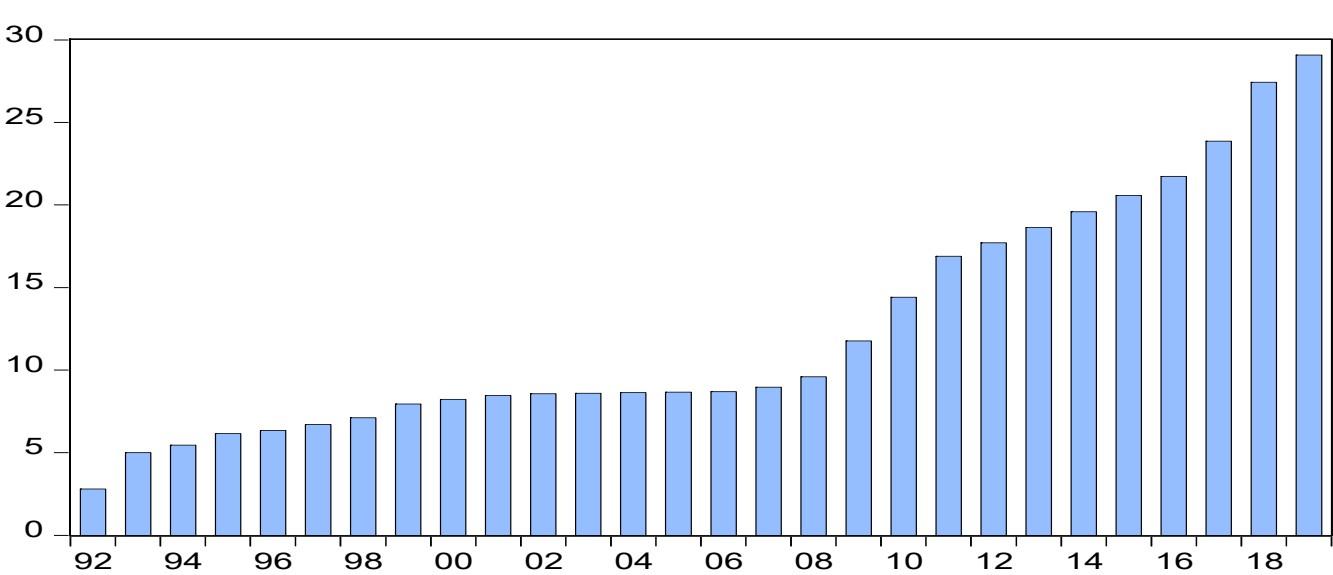

**Figure 3.** The movement of the exchange rate (ER) in Ethiopia. Source: Constructed by authors using E-view 10.

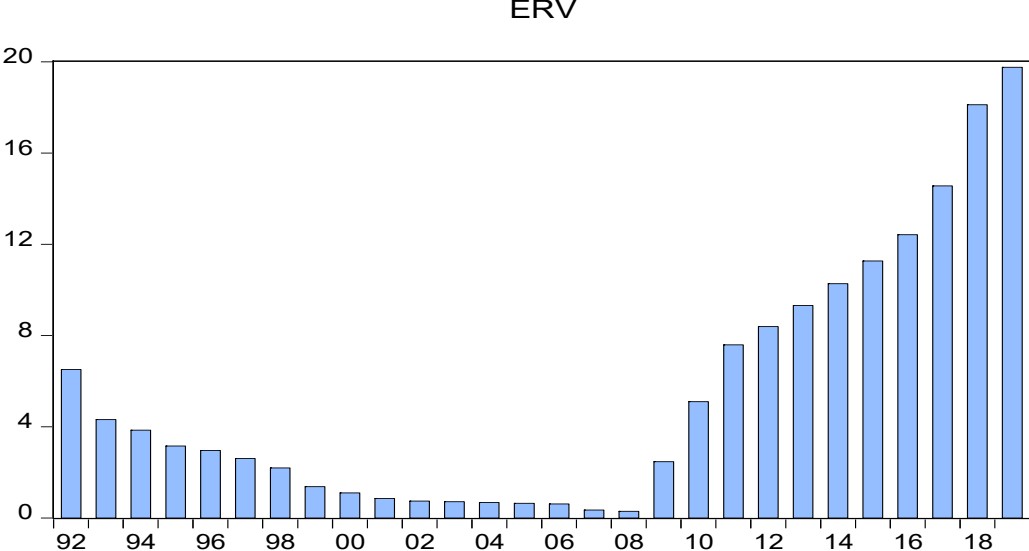

**Figure 4.** The movement of the exchange rate volatility (ERV) in Ethiopia. Source: Constructed by authors using E-view 10.

*4.2. Econometric Analysis*

To ascertain the goodness of fit of the estimated model, a diagnostic test was conducted. Accordingly, the diagnostic test suggested that the model passed the test of serial correlation, nonnormality of the errors, multicollinearity tests, heteroscedasticity associated with the model, and finally steadiness of the model, as presented in Appendix A.

$$TOP = -1.58 + 0.14EX + 0.225ERV + 0.24FDI + 0.016INF + 0.016GDP$$

4.2.1. Discussion of the Long-Term Estimation of Variables
Exchange Rate

According to the results of the regression analysis, the level of the exchange rate has a positive impact on international trade in Ethiopia. According to the coefficient on the variable, a 1% increase (devaluation) in domestic currency against foreign currency results in a 0.14 percent increase in Ethiopian international trade. The effect of this variable, however, was found to be insignificant. The justification for a positive relationship between the variables implies that the exchange rate influences the trade surplus or deficit, the reverse is true. In general, however, a weaker domestic currency stimulates exports while increasing the cost of imports. A strong domestic currency, on the other hand, hinders exports while making imports cheaper. As a result, devaluation is expected to improve a country's trade performance by increasing exports and decreasing imports. Furthermore, if the trend of export levels is declining, the relationship between the variables may be insignificant. The finding the study is consistent with the study established by (Parajuli 2012; Keho 2021). Furthermore, variables such as foreign direct investment, economic growth, and the inflation rate have a long-term positive and significant impact on international trade.

Exchange Rate Volatility

From the regression result, shown in Table 6, with other aspects remaining constant, the long-term analysis result shows that exchange rate volatility affects international trade positively and significantly. The coefficient on the exchange rate volatility variable was 0.225 percent. This implies that an increase in exchange rate volatility by 1 percent causes international trade to increase by 0.225 percent, and it was statically significant at a 1 percent level. The reason for the positive relationship is that in Ethiopia, the volatility of the exchange rate is slow, making it difficult for traders to make an immediate decision. With this in mind, it strongly encourages investors to enter the domestic market because

the volatility of the exchange rate is in one direction (devaluation), which would initiate ex-port-oriented traders. Furthermore, unlike developed countries, developing and least developed countries, such as Ethiopia, where forward exchange markets are in their infancy, are unable to diversify exchange rate volatility. Hence, it is the government's responsibility to follow up on the exchange rate movement to protect the traders. The result of this study is consistent with the finding established by (Asseery and Peel 1991), (De Grauwe and Decupere 1992), (Lothian and Taylor 1997), and (Bahmani-Oskooee and Hajilee 2009).

**Table 6.** Error Correction Model and short-term coefficients of the variables.

| ECM Regression | | | | |
|---|---|---|---|---|
| Case 2: Restricted Constant and No Trend | | | | |
| Variable | Coefficient | Std. Error | t-Statistic | Prob. |
| D(LER) | −0.608615 | 0.201375 | −3.022294 | 0.0106 |
| D(LERV) | 0.237194 | 0.033358 | 7.110514 | 0.0000 |
| D(LFDI) | 0.018455 | 0.011515 | 1.602692 | 0.1350 |
| D(INF) | 0.004909 | 0.000500 | 9.819086 | 0.0000 |
| D(INF(-1)) | −0.002574 | 0.000450 | −5.722758 | 0.0001 |
| D(GDP_RATE) | 0.004073 | 0.000945 | 4.310824 | 0.0010 |
| D(GDP_RATE(-1)) | −0.002449 | 0.000696 | −3.517468 | 0.0042 |
| CointEq(-1) * | −0.556430 | 0.045855 | −12.13449 | 0.0000 |
| R-squared | 0.895840 | Mean dependent var | | 0.051206 |
| Adjusted R-squared | 0.855333 | S.D. dependent var | | 0.047361 |
| S.E. of regression | 0.018014 | Akaike info criterion | | −4.947692 |
| Sum squared resid | 0.005841 | Schwarz criterion | | −4.560585 |
| Log likelihood | 72.32000 | Hannan-Quinn criter. | | −4.836219 |
| Durbin-Watson stat | 2.640991 | | | |

Source: Generated by authors from E-views 10.

Control Variables

As depicted by the regression analysis, economic growth proxies by GDP growth rate have a positive and significant effect on international trade in Ethiopia. An increase in GDP is expected to enhance the performance of a country in the international market. It implies that the overall economic development is moving on an upward scale, which includes the growth of exports. The growth of exports opens the way for FDI. Hence, GDP has an unexpected relationship with international trade, and FDI has the expected direction of relationship with the dependent variable in the long run. Some empirical evidence reveals that there is a negative relationship between exchange rate volatility and economic growth in developed and industrial countries (Vieira et al. 2013), (Janus and Riera-Crichton 2015), (Papadamou et al. 2016), middle-income countries (Aizenman et al. 2018), as well as developing ones (Dollar 1992). Other studies by (Bleaney and Greenaway 2001) in some sub-Saharan African countries show volatility exerts negative effects on investment but not on economic growth. Another study by (Han 2020) found that exchange rate fluctuation had different effects on economic growth in different countries.

The other variable used in this study as a control variable was inflation. The result of regression analysis implied that inflation has a positive and significant effect on international trade in Ethiopia. Domestically, rising inflation makes local goods more expensive and less attractive to customers at home, who increasingly turn to cheaper imports. In this case, the customers decision is dependent on government trade strategies. These higher prices can also reduce exports because of competition in international trade.

After accepting the long-term analysis results, the short-term error correction model was estimated. The Error Correction Model (ECM) indicates the speed of adjustment to restore equilibrium in the dynamic model. From the above error correction model (Table 6), the mark of the ECM term gives the error correction coefficient, which is negative and statistically significant as expected. This means that the adjustment speed is good. According to (Bahmani-Oskooee and Hajilee 2009) and (Bahmani-Oskooee and Satawatananon 2007),

if ECM is negative and statistically significant, it means the variables are cointegrated in the long term. The ECM, which indicates the speed of adjustment, has a value of $-0.556430$. It is considered correctly signed and also statistically significant. This implies that the short-term disequilibrium, as well as inconsistencies, are adjusted and corrected in the long run at a percentage of 55.6%. The negative sign is a confirmation of the existence of equilibrium in long term.

### 4.2.2. Discussion of Short-Term Effects

As shown in Table 6, for the short-term coefficients of the variables, it can be seen that the variable exchange rate and international trade (trade openness) have a negative relationship, which is statistically significant at 5 percent. This implies that depreciation raises the cost of imports and has a short-term impact on international trade. In the short term, exchange rate volatility is still positive and has a significant impact on international trade. This result is heavily influenced by the direction of Ethiopian exchange rate volatility, which is primarily on the devaluation side. The finding of this study is consistent with the study established by (Thuy and Thuy 2019). As with the long-term estimation, the sign and direction of all control variables were found the same in the short-term as of long term.

### 5. Conclusions

The study aimed to examine the effect of the exchange rate level and its volatility on international trade in Ethiopia during the study period of 1992–2019. The standard deviation of the exchange rate was used as a measure of exchange rate volatility and the Autoregressive Distributed Lag (ARDL) model was used to investigate the relationship between variables. In addition to the exchange rate level and its volatility, the study included FDI, economic growth, and inflation as control variables of the study. The study examined both long-term and short-term relationships between variables, and the finding of the study implies that the exchange rate level found positively affects international trade but insignificantly in long run. However, in the short run, it affects negatively and significantly. This finding lends support to the J-curve effects, which suggest an initial loss in short run followed by a dramatic gain in the long run. However, the findings of this study suggest that there is no significant gain from international trade to justify currency depreciation in Ethiopia.

In addition, exchange rate volatility has a positive and significant effect in both the short term and long-term estimation. Furthermore, FDI, economic growth, and inflation were found to positively and significantly affect international trade movement in both the long-term and short-term estimation. To run the model, the study was limited to only 28 years of annual data, which were obtained from 1992 to 2019. This is because Ethiopia had not implemented a floating exchange rate before 1992. Furthermore, the study would be more meaningful if the collected data were analyzed quarterly or semiannually to capture the relationship between the dependent and independent variables. However, due to the scarcity of such data, the researchers were forced to investigate the relationship between variables on a yearly basis. As a result, the researchers advise other scholars to find a way to collect data and examine the relationship to determine whether the effect of the J-curve is insignificant in Ethiopia.

**Author Contributions:** Conceptualization, original draft preparation, T.N.; investigation, resources, and data curation, B.O.; methodology, analysis and software, G.D.; validation, and supervision, A.T.; project administration, and funding acquisition, M.F.-F. All authors have read and agreed to the published version of the manuscript.

**Funding:** The APC was funded by Hungarian University of Agriculture and Life Science, Doctoral school of Economic and regional sciences.

**Institutional Review Board Statement:** Not applicable.

**Informed Consent Statement:** Not applicable.

**Data Availability Statement:** The data can be available based on the request@goshudasalegn@gmail.com or (https://data.worldbank.org/country/ET, accessed on 1 September 2021) and (https://nbebank.com/, accessed on 1 September 2021).

**Conflicts of Interest:** The authors declare no conflict of interest.

## Appendix A

**Table A1.** Multicollinearity test result.

| Correlation | LER | LERV | INF | GDP_RATE | LFDI |
|---|---|---|---|---|---|
| LER | 1.000000 | | | | |
| LERV | 0.551352 | 1.000000 | | | |
| INF | 0.273061 | −0.048720 | 1.000000 | | |
| GDP_RATE | 0.450009 | 0.020344 | 0.109587 | 1.000000 | |
| LFDI | 0.766439 | 0.230563 | 0.112199 | 0.400018 | 1.000000 |

Variance Inflation Factors
Date: 04/27/21 Time: 07:16
Sample: 1992 2019
Included observations: 26

| Variable | Coefficient Variance | Uncentered VIF | Centered VIF |
|---|---|---|---|
| LTOP(-1) | 0.013495 | 710.3265 | 110.9674 |
| LER | 0.187119 | 11702.36 | 468.4600 |
| LER(-1) | 0.147517 | 8724.664 | 352.7514 |
| LERV | 0.003604 | 95.82862 | 59.36351 |
| LERV(-1) | 0.002504 | 60.01078 | 37.55093 |
| LFDI | 0.000551 | 2167.060 | 13.49321 |
| LFDI(-1) | 0.000443 | 1702.982 | 13.61822 |
| INF | $4.92 \times 10^{-7}$ | 5.855531 | 3.118005 |
| INF(-1) | $9.47 \times 10^{-7}$ | 10.80057 | 6.011297 |
| INF(-2) | $4.86 \times 10^{-7}$ | 5.458318 | 3.066066 |
| GDP_RATE | $2.75 \times 10^{-6}$ | 11.92969 | 2.754051 |
| GDP_RATE(-1) | $2.49 \times 10^{-6}$ | 11.32760 | 2.628771 |
| GDP_RATE(-2) | $1.28 \times 10^{-6}$ | 5.870233 | 2.051939 |
| C | 0.056955 | 3042.277 | NA |

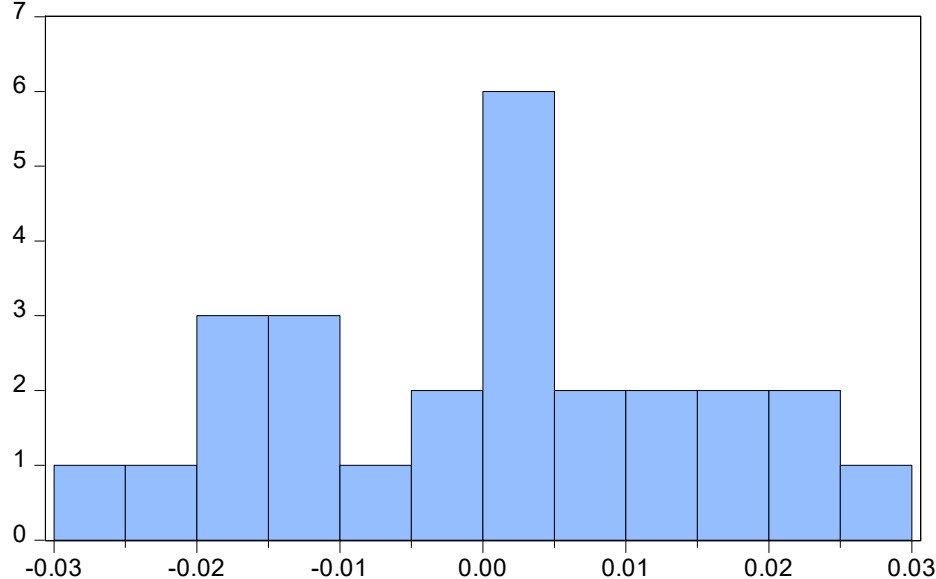

**Figure A1.** Normality test result.

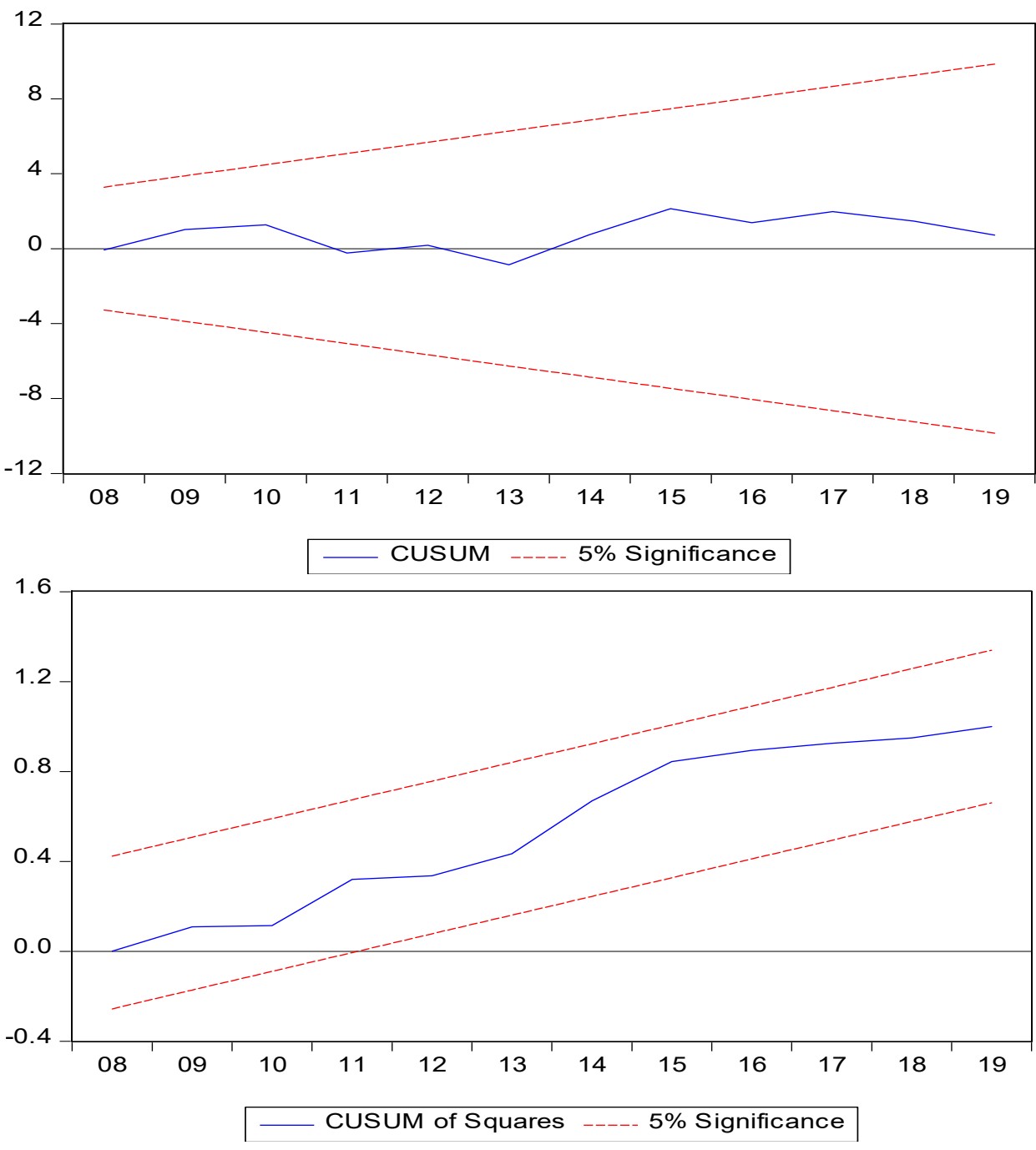

**Figure A2.** Model Stability test result.

**Table A2.** Serial correlation test result.

| Breusch-Godfrey Serial Correlation LM Test: | | | |
|---|---|---|---|
| F-statistic | 1.232675 | Prob. F(2,10) | 0.3323 |
| Obs*R-squared | 5.142183 | Prob. Chi-Square(2) | 0.0765 |

**Table A3.** Heteroscedasticity test result.

| Heteroscedasticity Test: Breusch-Pagan-Godfrey | | | |
|---|---|---|---|
| F-statistic | 1.092469 | Prob. F(13,12) | 0.4422 |
| Obs*R-squared | 14.09256 | Prob. Chi-Square(13) | 0.3674 |
| Scaled explained SS | 1.671491 | Prob. Chi-Square(13) | 0.9999 |

**Table A4.** Model Specification test result.

| Ramsey RESET Test | | | |
|---|---|---|---|
| Equation: UNTITLED | | | |
| Specification: LTOP LTOP(-1) LER LER(-1) LERV LERV(-1) LFDI LFDI(-1) INF INF(-1) INF(-2) GDP_RATE GDP_RATE(-1) GDP_RATE(-2) C | | | |
| Omitted Variables: Squares of fitted values | | | |
| | Value | df | Probability |
| t-statistic | 0.559799 | 11 | 0.5868 |
| F-statistic | 0.313375 | (1,11) | 0.5868 |

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
