# Peer review of "Does the Exchange Rate and Its Volatility Matter for International Trade in Ethiopia?"

_jrfm, doi:10.3390/jrfm14120591_

Round 1

Reviewer 1 Report

The paper focuses on a contemporary and actual topic and investigating a remarkable correlation between exchange rate volatility and international trade via the case of Ethiopia. The title reflects the content well, the abstract is well written, the keywords are appropriate. The introduction should be improved, highlighting the context, limitations, and relevance of research goals better.

The literature review is very short, it should be extended, involving and critically analyzing more, recent international sources, preferably from Scopus/WoS listed journals.

The methodology is demonstrated well, detailed enough, and comprehensive. The results are clear, supported by the methodological toolset, organized along with the hypotheses.

The conclusions are clear, well understandable, and relevant.

Author Response

Based on the comment provided, we have corrected.

Reviewer 2 Report

1 - Abstract:

the following two sentences seem contradictory: "In the short run, the exchange rate level is found to negatively and significantly influence international trade. However, exchange rate volatility founds positively and significantly affects international trade in the short run." Considering the contents of the paper in the second sentence you should correct "in the short run", with "in the long run"

2 - rows 42- 43:

ERRATA: "Since 1990, numerous studies by different scholars such as (Thorbecke, W., & Kato, 2012), Mbam & Michael (2020), and (Oloyede & Essi, 2017) ..."

CORRIGE: "Since 1990, numerous studies by different scholars such as Thorbecke, W., & Kato, (2012), Mbam & Michael (2020), and Oloyede & Essi (2017) ..."

3 - The literature references on the effects of exchange rate changes on international trade need to be expanded. Among the most relevant studies that cannot fail to be mentioned are: 

Bahmani-Oskooee, M., and T. J. Brooks. 1999. “Bilateral J-Curve between U.S. and Her Trading Partners.” Weltwirtschaftliches Archiv 135: 156–165. doi:10.1007/ BF02708163.

Bahmani-Oskooee M. and Ratha A. (2004), “The J-Curve: A Literature Review”, Journal of Applied Economics, 36 (13),  pp. 1377-1398.

Bahmani-Oskooee, M., and Y. Wang. 2008. “The J-Curve: Evidence from Commodity Trade between U.S. and China.” Applied Economics 40: 2735–2747. doi:10.1080/ 00036840600970328.

Lucarelli S., Andrini F.U. and Bianchi A. (2018), “Euro Depreciation and Trade Asymmetries between Germany and Italy versus the US: Industry-level Estimates”, Applied Economics, 50 (1), pp. 15-34.

Gobbi L. and Lucarelli S. (2021), "ECB quantitative easing, euro depreciation and supply chains: Industry-level estimates for Germany, Italy and Greece. New prospects for a Minskyan big bank?", PSL Quarterly Review, 74 (296), pp. 25-50

In particular, the contributions of Lucarelli, Andrini, and Bianchi (2018) and Gobbi and Lucarelli (2021) point out that looking at different industries even within the same country there can be different effects of currency depreciation on international trade.

4 -  row 52: Delete the following part of the sentence "section one is the introduction of the research"

5 - Literature Survey

The part of the paragraph  from row 58 to row 74 seems overly didactic to me. I believe that this paragraph could begin directly with the part of the text that is now on line 75 ("Exports of agricultural products are ..."). 

The following part of the paragraph should better review the existing literature specifically devoted to: 

i) the effects of the exchange rate variation on international trade in Africa. See:
Bahmani-Oskooee, M., Gelan, A. (2012). Is There J-Curve Effect in Africa? International Review of Applied Economics, 26 (1), pp. 73-81. 
Chebbi, H., E., Olarreaga, M. (2019). Investigating Exchange Rate Shocks on Agricultural Trade Balance: The Case of Tunisia. The Journal of International Trade and Economic Development, 28 (5), pp. 628-647. 
Yaya Keho | (2021) Determinants of Trade Balance in West African Economicand Monetary Union (WAEMU): Evidence from heterogeneous panel analysis, Cogent Economics& Finance, 9:1, 1970870, DOI: 10.1080/23322039.2021.1970870

(ii) The effects of the exchange rate variation in the agricultural sector. See: 
Batten, D. S., Belongia, M. T. (1986). Monetary Policy, Real Exchange Rates, and U.S. Agricultural Exports. American Journal of Agricultural Economics, 68 (2), pp. 422-427. 
Baek, J., Koo, W. W., Mulik, K. (2009). Exchange Rate Dynamics and the Bilateral Trade Balance: The Case of US Agriculture. Agricultural and Resource Economics Review, 38 (2), pp. 213-228.

It would then be helpful to the reader if some references were given to official documents of the Central Bank of Ethiopia showing the views of this institution on the expected effects of devaluation in Ethiopia.

5- References to the sources from which the data used for the estimates are drawn should be included.

6 - I recommend to perform as a diagnostic also a RESET test, in order to tests whether non-linear combinations of the fitted values help explain the response variable. 

7 - It would be useful to have some information on Ethiopia's production specialization and the type of goods and services that are mostly exported and mostly imported.

8 - The statistical nonsignificance found in the results depends on the small number of available observations (27 years).
I do not know the state of statistics available in Ethiopia, (checking on the site of the FED in St. Louis and on that of the World Bank I note that there are only annual data available); perhaps it is appropriate to indicate at the end of the article the need to make available to scholars data with a greater level of detail (at least quarterly). In this way, more reliable estimates could be developed, using the ratio between exports and imports as the dependent variable, as is the case in the most relevant literature on the issues discussed in the article.

9 - The fact that you find negative effects in the short term and positive (although not significant) effects in the long term makes your study fit into the J curve literature. I suggest you discuss it, even though the J curve literature uses trade balance as the dependent variable and in your study you use openness to international trade instead.

Author Response

Based on the comment provided, we have corrected 

Reviewer 3 Report

I think that this is an interesting article. I hope that my comments will be of some help for the authors to improve the article. 

(1) Relationship to Literature: Literature has not been extensively considered. The paper demonstrates an inadequate understanding of the relevant literature in the field and thus an appropriate range of literature sources and significant works are ignored.

(2) Conclusion: The results need expansion — The results should be discussed and compared with other studies. Especially, the authors should clearly explain the similarity and differences between their results and those of previous research.

(3) Implications for research, practice and/or society: The paper must identify clearly any implications for research, practice and/or society. Thus, the paper must bridge the gap between theory and practice. How can the research be used in practice (economic and commercial impact), to influence public policy, in research? What is the impact on society? Are these implications consistent with the findings and conclusions of the paper?

Author Response

based on the comment provided, we have corrected.

Round 2

Reviewer 2 Report

1. In my previous report I wrote that the following two sentences seem contradictory: "In the short run, the exchange rate level is found to negatively and significantly influence international trade. However, exchange rate volatility founds positively and significantly affects international trade in the short run." Considering the contents of the paper in the second sentence you should correct "in the short run", with "in the long run"

In you aswers you wrote: "As stated, the sentence is not contradict each other. We used two variables (Exchange rate, and exchange rate volatility). The result of both variable is different in the short run. Exchange rate- negative and significant. Exchange rate volatility-positive and significant"

Reading again the paper I can read that you find both a positive and significant impact on the international trade both in the short and in the long run (see row 370 and paragraph 3.2.1). Probably the best way to present your results in the abstract is: "In the short run, the exchange rate level is found to negatively and significantly influence international trade. However, exchange rate volatility founds positively and significantly affects international trade both in the short and in the long run."

2.  As I reccomend in my previous report, the literature references on the effects of exchange rate changes on international trade need to be expanded. Among the most relevant studies that cannot fail to be mentioned, in your section 2, are: 

Bahmani-Oskooee, M., and T. J. Brooks. 1999. “Bilateral J-Curve between U.S. and Her Trading Partners.” Weltwirtschaftliches Archiv 135: 156–165. doi:10.1007/ BF02708163.

Bahmani-Oskooee M. and Ratha A. (2004), “The J-Curve: A Literature Review”, Journal of Applied Economics, 36 (13),  pp. 1377-1398.

Bahmani-Oskooee, M., and Y. Wang. 2008. “The J-Curve: Evidence from Commodity Trade between U.S. and China.” Applied Economics 40: 2735–2747. doi:10.1080/ 00036840600970328.

Lucarelli S., Andrini F.U. and Bianchi A. (2018), “Euro Depreciation and Trade Asymmetries between Germany and Italy versus the US: Industry-level Estimates”, Applied Economics, 50 (1), pp. 15-34.

Gobbi L. and Lucarelli S. (2021), "ECB quantitative easing, euro depreciation and supply chains: Industry-level estimates for Germany, Italy and Greece. New prospects for a Minskyan big bank?", PSL Quarterly Review, 74 (296), pp. 25-50

In particular, the contributions of Lucarelli, Andrini, and Bianchi (2018) and Gobbi and Lucarelli (2021) point out that looking at different industries even within the same country there can be different effects of currency depreciation on international trade.

3. The results of the diagnostic tests presented in the appendix should be commented on in the text of the article at the end of section 3.

Typos

row 61: ERRATA: furthermore. Evidence; CORRIGE: Furthermore, evidence

row 310: ERRATA: Exchange rate; CORRIGE: exchange rate

The right number of the conclusive section is not 3, it is 4!

Author Response

We would like to thank you for your contribution to the article improvement.

Reviewer 3 Report

I think that this article has been we4ll revised. I am glad to suggest that this article can be accepted for publication.

Author Response

(The authors gave the same response as above.)

Round 3

Reviewer 2 Report

I suggest to accept the paper for the publication.